# Effects of Pueraria Extract and Curcumin on Growth Performance, Antioxidant Status and Intestinal Integrity of Broiler Chickens

**DOI:** 10.3390/ani13081276

**Published:** 2023-04-07

**Authors:** Shuangshuang Guo, Jinchao Hu, Sihan Ai, Lanlan Li, Binying Ding, Di Zhao, Lei Wang, Yongqing Hou

**Affiliations:** Engineering Research Center of Feed Protein Resources on Agricultural By-Products, Ministry of Education, Hubei Key Laboratory of Animal Nutrition and Feed Science, Wuhan Polytechnic University, Wuhan 430023, China

**Keywords:** pueratia extract, curcumin, antioxidant status, intestinal integrity, broiler chicken

## Abstract

**Simple Summary:**

Plant extracts are one of the alternatives to antibiotics, and are generally considered to be safe for animals and effective against pathogens. Their antimicrobial, anti-inflammatory, antioxidant, and antiviral activities have been well documented. Puerarin, the main component of pueraria extract, is a C-glucoside of isoflavone daidzein. Curcumin, isolated from the rhizome of curcuma, is a kind of polyphenol. The current study was carried out to examine the synergistic effects of pueraria extract and curcumin on the growth performance, antioxidant capacity and intestinal barriers of broilers. Our observations showed that supplementation of pueraria extract and curcumin alone or in combination did not improve the growth performance of broilers in the 28-day trial, but enhanced the antioxidant status and intestinal integrity of broilers by increasing the activities of antioxidant enzymes and promoting intestinal morphology. Pueraria extract and curcumin are potential modulators of antioxidant function and intestinal health. Their beneficial effects on improving growth performance need further investigation.

**Abstract:**

The current study was carried out to examine the effects of pueraria extract (PE) and curcumin (CUR) on growth performance, antioxidant capacity and intestinal integrity in broiler chickens. A complete randomized design with a 2 × 2 factorial arrangement of treatments was employed to assign 200 one-day-old Ross-308 broilers to four groups, each including five replicates of ten birds. Chickens in the control group (CON) were fed the basal diet, while the PE, CUR and PE+CUR groups were supplemented with 200 mg/kg PE or 200 mg/kg CUR or 200 mg/kg PE+ 200 mg/kg CUR. This trial lasted for 28 days. The PE supplementation decreased the average daily gain during the whole period (*p* < 0.05). The PE+CUR group had a higher feed conversion ratio than that of the PE and CUR groups during days 14–28 and 1–28 (*p* < 0.05). Dietary CUR supplementation increased duodenal T-SOD activity (*p* < 0.05). Compared with the CON group, the other three groups increased the duodenal GSH-Px activity, the PE+CUR group reduced the duodenal H_2_O_2_ level, and the CUR and PE groups elevated the ileal GSH-Px activity and the ratio of jejunal villus height to crypt depth, respectively (*p* < 0.05). The addition of PE decreased crypt depth and increased villus area and mucin-2 mRNA level in the jejunum (*p* < 0.05). Overall, dietary supplementation with PE, CUR, or a combination of these, enhanced the antioxidant status and intestinal integrity of broilers.

## 1. Introduction

For the last several decades, antibiotics have been extensively used in feeds for the improvement of animal growth and prevention of diseases [1]. However, high and long-term addition of antibiotics to feed causes growing antimicrobial resistance in bacteria, which poses a great risk to human health [2]. Therefore, the use of antibiotics as feed additives has been restricted by increasing legal regulations in the United States, the European Union, China and many other countries. Plant extracts are one of alternatives to antibiotics, and are generally considered to be safe for animals and effective against certain bacteria. Plant extracts are extensively used as growth promoters and health protectants in feeds, due to their antimicrobial, anti-inflammatory, antioxidant, and antiviral activities [3]. Pliego et al. [4] reported that encapsulated plant extracts or their metabolites are potential alternatives to antibiotic growth promoters in poultry nutrition in the future.

The dried root of pueraria, also known as Kudzu root (Gegen in Chinese), is commonly used in traditional Chinese medicine [5]. Puerarin (PR) is a C-glucoside of the isoflavone daidzein extracted from pueraria [6]. The PR not only plays an anti-inflammatory role by downregulating the expression of proinflammatory cytokines or inhibiting the production of inflammatory mediators [7,8], but also plays an antioxidant role, by enhancing the activity of antioxidant enzymes in serum and tissues [9]. Niu et al. [10] reported that PR treatment protected the lung from *Mycoplasma gallisepticum* infection-induced damage by inhibiting the inflammatory response and apoptosis in chickens. Curcumin (CUR) is a polyphenolic compound isolated from the rhizome of curcuma, which has various biological functions such as antibacterial, anti-inflammatory, antioxidant and antiviral [11]. Therefore, CUR, as a purely natural and non-polluting functional feed additive, has good prospects for application in livestock and poultry production. Yadav et al. [12] demonstrated that CUR exhibited some positive effects on antioxidant capacity, lesion score and oocyst shedding in broilers challenged with *Eimeria* species.

The synergistic effects of PR and CUR have scarcely been investigated in animal production. However, in mammals, gold-nanoparticle-formulated PR and CUR effectively suppressed the lipopolysaccharide (LPS)-induced inflammation and cytotoxicity in rats [13]. Singh et al. [14] reported that the combination of PR and CUR suppressed the drinking addiction-related and inflammation-related abnormalities, using P-rats as a model. In general, the combination of the two plant extracts in livestock feeds can avoid the limitations and high cost of single functional substances, thus increasing the economic efficiency of livestock production. Therefore, this study investigated the effects of pueraria extract (PE) and CUR, alone or in combination, on growth performance, antioxidant status and intestinal integrity in broiler chickens, and explored the interaction between PE and CUR, to provide a reference for further screening of alternatives to antibiotics.

## 2. Materials and Methods

### 2.1. Experimental Design, Birds and Management

In the present study, all animal procedures obtained approval from the Institutional Animal Care and Use Committee of Wuhan Polytechnic University (protocol code: WPU202108001). A 2 × 2 factorial randomized complete block design was used to investigate the main and interactive effects of pueraria extract (PE) and CUR, both of which were supplemented at 0 and 200 mg/kg of feed. The 200 1-day-old Ross 308 broiler chicks obtained from a commercial hatchery, with similar initial body weights, were randomly assigned to four groups. There were 5 replicates of 10 chicks (5 males and 5 females) in each group. Birds in the control group (CON) were fed a corn–soybean meal basal diet. Birds in the PE and CUR groups were fed basal diets supplemented with 200 mg/kg PE and 200 mg/kg CUR, respectively. Birds in the PE+CUR group were fed a basal diet supplemented with both 200 mg/kg PE and CUR. The basal diet was formulated according to the NY/T33-2004 recommendations for broilers, and the composition and nutrient levels of the basal diet are shown in Table 1. The PE containing 238 g/kg puerarin was provided by Hubei Horwath Biotechnology Co., Ltd. (Wuhan, China). The CUR with a purity of 98% was purchased from Macklin Inc. (Shanghai, China). The trial lasted for 28 days. All broilers had free access to feed and water in a room where the temperature was manually controlled. Room temperature was controlled at 32~34 °C for the first week and then reduced by 2~3 °C per week to a final temperature of 24 °C, and remained constant thereafter. The relative humidity of the chicken house was controlled at 50~60% throughout the trial.

### 2.2. Sample Collection

At 28 days of age, two chickens (one male and one female) of near to the average pen body weight (1.11~1.19 kg) were selected from each replicate. Blood samples were collected from wing veins using heparinized anticoagulant vacuum blood-collection tubes. Blood samples were centrifuged at a relative centrifugal force of 1710 g for 10 min at 4 °C to separate the plasma, which was then stored at −80 °C until analysis. Selected chickens were intravenously injected with sodium pentobarbital at 50 mg/kg of body weight. Then, the birds were slaughtered by cervical dislocation. About 1 cm of intestinal segment was sampled from the middle of the duodenum (between the gizzard outlet and the entry of the bile and pancreatic ducts), the jejunum (between the end of the duodenum and Meckel’s diverticulum), and the ileum (between Meckel’s diverticulum and the ileocecal junction). The intestinal segments were immediately fixed in 4% paraformaldehyde, for morphological measurements. The remaining intestinal samples from the duodenum, jejunum and ileum were collected and stored at −80 °C until analysis.

### 2.3. Growth Performance

Broilers in each replicate were weighed at 14 and 28 days of age, and feed consumption was recorded for each replicate at days 1–14, 14–28 and 1–28. The average daily gain (ADG), average daily feed intake (ADFI) and mortality-adjusted feed conversion ratio (FCR) were calculated for each replicate.

### 2.4. Antioxidant Status

Approximately 1 g of liver, duodenum, jejunum and ileum sample were homogenized in 9 mL of 0.9% ice-cold saline and centrifuged at relative centrifugal force of 1710 g for 10 min at 4 °C. The supernatant was collected for antioxidant status analysis. The levels of malondialdehyde (MDA) and hydrogen peroxide (H_2_O_2_), and the activities of total antioxidant capacity (T-AOC), total superoxide dismutase (T-SOD), catalase (CAT), and glutathione peroxidase (GSH-Px) in the plasma and supernatants were determined using commercial assay kits (Nanjing Jiancheng Institute of Bioengineering, Nanjing, China), according to the manufacturer’s instructions [15].

### 2.5. Histological Mesurements of Small Intestine

The fixed intestinal samples (duodenum, jejunum and ileum) were first washed under running water, subsequently dehydrated in gradient concentrations of ethanol (50%, 75%, 85%, 90%, 95% and 100% ethanol), and then embedded in paraffin. Approximately 4-µm intestinal cross-sections were collected on glass slides and finally stained with hematoxylin and eosin for morphological measurements. The images were captured with a light microscope (OLYMPSBX-41TF). Ten intact intestinal villi in each section were measured. Villus height (VH, between the tip of the villi and the villus–crypt junction), crypt depth (CD, between this junction and the base of the crypt) and villus area (VA) were measured using an ImagePro Plus software (Media Cybernetics), as previously reported [1]. The ratio of VH to CD (VH/CD) was calculated.

### 2.6. RNA Extraction and Quantitative Analysis of Tight Junction mRNA with Real-Time PCR

The expressions of zonula occludens-1 (ZO-1), claudin-1, occludin, and mucin-2 mRNA in the jejunum were determined using a real-time PCR system (ABI 7500; Applied Biosystems, Foster City, CA, USA) following the protocol of SYBR^®^ Premix Ex TaqTM kit (Takara, Dalian, China). First, the total RNA was extracted from the jejunum using a Trizol reagent (Invitrogen, Grand Island, NY, USA), according to the manufacturer’s instructions. The concentration and purity of total RNA were quantified by measuring its optical density at 260, 230 and 280 nm with a NanoDrop^®^ ND-2000 UV-VIS spectrophotometer (Thermo Scientific, Waltham, MA, USA). Reverse transcription was performed from 1 μg of total RNA using the PrimeScript^®^ RT reagent kit with gDNA Eraser (Takara, Dalian, China), according to the manufacturer’s instructions. The primer sequences of target and housekeeping genes are listed in Table 2. A PCR program consisted of initial denaturation at 95 °C for 30 s, 40 cycles at 95 °C for 5 s, and annealing and extension at 60 °C for 34 s was employed. Samples were run in triplicate. The housekeeping gene of β-actin was used, and relative mRNA levels were calculated using the 2^−ΔΔCt^ method [16].

### 2.7. Statistical Analysis

Data were analyzed using a two-way ANOVA, using a univariate general linear model with SPSS 26.0 (SPSS Inc., Chicago, IL, USA). All data complied with normal distribution and variance homogeneity, which was confirmed before the data analysis. In cases where the interactive effects between PE and CUR were significant, the means were compared using Duncan’s multiple test. Statistical significance was considered as *p* < 0.05. Data were expressed as means and SEM.

## 3. Results

### 3.1. Growth Performance

As shown in Table 3, PE supplementation decreased ADG in broilers at 1–14 days, 14–28 days and 1–28 days (*p* < 0.05). At 14–28 days and 1–28 days, PE and CUR showed an interactive effect on ADFI and FCR (*p* < 0.05). At 14–28 days, ADFI and FCR were lower in the CON and PE+CUR groups than in the PE and CUR groups (*p* < 0.05). At 1–28 days, ADFI was lower in the PE+CUR group than that in the PE and CUR groups (*p* < 0.05), and the ADFI in the CON group was not significantly different from the other three groups (*p* > 0.05). FCR was significantly increased in the PE and CUR groups compared to the CON group, at 1–28 days of age (*p* < 0.05).

### 3.2. Antioxidant Status

As seen in Table 4, PE and CUR exhibited interactive effects on plasma GSH-Px and CAT activities (*p* < 0.05). Plasma GSH-Px activity was significantly different among the four groups in the order PE > CUR > PE+CUR > CON (*p* < 0.05). Plasma CAT activity was significantly lower in the PE and CUR groups compared to the CON group (*p* < 0.05). Both PE and CUR decreased GSH-Px activity in the liver (*p* < 0.05). Dietary CUR supplementation increased T-SOD activity in the duodenum (*p* < 0.05, Table 5). There was an interactive effect of PE and CUR on GSH-Px activity in the duodenum and ileum and on H_2_O_2_ content in the duodenum (*p* < 0.05). Compared with the CON group, the other three groups increased duodenal GSH-Px activity, the PE+CUR group reduced duodenal H_2_O_2_ content, and the CUR group elevated ileal GSH-Px activity (*p* < 0.05). The antioxidant status of the jejunum was not significantly affected by PE and CUR supplementation (*p* > 0.05).

### 3.3. Intestinal Morphology

The images of intestinal morphology are presented in Figure 1. Morphological evaluation showed that CUR supplementation significantly increased VH/CD in the duodenum (*p* < 0.05, Table 6). The PE supplementation decreased CD and increased VA in the jejunum (*p* < 0.05). There was an interactive effect of PE and CUR on VH/CD in the jejunum (*p* < 0.05). Jejunal VH/CD was higher in the PE group than in the CON group (*p* < 0.05). Ileal morphology was not significantly affected by PE and CUR (*p* > 0.05).

### 3.4. Relative mRNA Levels of Claudin-1, Occludin, ZO-1 and Mucin-2 in Jejunum

As shown in Table 7, PE supplementation upregulated the mRNA level of mucin-2 in the jejunum (*p* < 0.05). Supplementation of PE and CUR had no significant effect on the relative mRNA levels of claudin-1, occludin and ZO-1 in the jejunum (*p* > 0.05).

## 4. Discussion

During poultry production, the environment of poultry house and feed contamination can cause oxidative stress in poultry, resulting in oxidative damage. Oxidative damage can lead to a significant decrease in the quality of animal products, and seriously affect intestinal health [17]. Both PE and CUR are compounds extracted from natural plants. Although their structures and physiological functions are different, both PE and CUR have growth-promoting, antibacterial, anti-inflammatory, antioxidant, immunity-boosting and intestinal-flora-regulating functions [7,8,11]. The combined addition of PE and CUR in livestock and poultry production has been rarely reported, so further experimental studies are needed to determine how PE and CUR can be scientifically utilized in diets, and whether they can effectively replace antibiotics.

In the present study, PE decreased ADG throughout the experiment period, which is consistent with Payne et al. [18], who found that soy isoflavones reduced ADG and ADFI in commercial broilers. However, PR orally administered from days 8 to 18 recovered the impaired growth performance of broilers fed with thiram during days 4 to 7, to induce experimental tibial dyschondroplasia [19]. It was reported that 250 mg/kg CUR supplementation increased body weight, ADG and ADFI in genotype AA commercial broilers at 21 and 42 days of age, and significantly reduced FCR [20]. The CUR supplementation at 300 mg/kg increased ADG during days 1–21, regardless of LPS stimulation [21]. In our study, CUR supplementation had no significant effect on growth performance, and neither PE nor CUR addition had a significant effect on ADFI and FCR. The reason for the results may be due to different breeds of broilers, different growth stages, different diets, and different doses of PE and CUR in broiler diets.

The main endogenous antioxidant enzymes in the body system, including SOD, GSH-Px and T-AOC, were the main compounds preventing oxidative stress and cell damage [22]. The SOD formed the first defense line in cells against the negative effects of ROS by catalyzing the dismutation of endogenous superoxide radicals to H_2_O_2_, which is then eliminated by CAT and GSH-Px [23]. Additionally, MDA is a product of lipid peroxidation, indicating oxidative damage to lipids [24]. It is well known that PR has potential antioxidant activities both in vitro and in vivo [9], and that CUR can scavenge free radicals and stimulate antioxidant parameters [25,26]. PE has been shown to have a superior effect on enhancing the antioxidant status in C57Bl/6J mice in vivo, including increasing the activities of SOD and GSH-Px, and the contents of T-AOC and MDA [9]. In the present study, PE and CUR had interactive effects on GSH-Px activities in the plasma and duodenum and H_2_O_2_ content in the duodenum. GSH-Px activities in the plasma and duodenum were increased in broilers fed PE, and T-SOD activity was increased and H_2_O_2_ content in the duodenum was decreased in broilers fed CUR. Moreover, both PE and CUR decreased GSH-Px activity in the liver. Similar to our findings, Waqas et al. [19] revealed that PR treatment following thiram addition increased hepatic GSH-Px and T-SOD, and reduced MDA content in broilers. Zhang et al. [27] showed that the activities of GSH-Px, GST, CAT and T-SOD were significantly increased in the breast muscle of the CUR-fed compared to the heat-stressed group. This may be due to the fact that polyphenolic compounds are nitrosative reaction inhibitors that prevent oxidative damage via scavenging reactive oxygen species and enhancing SOD and GSH-Px activities [28].

In the present study, PE and CUR exhibited some positive responses regarding antioxidant capacity, suggesting that PE and CUR could be a dietary strategy to improve gut health in broilers. Therefore, the morphological structure of the small intestine of broilers fed with PE- and CUR-treated diets was measured. Normal small-intestinal histology is essential for good intestinal function [1]. It is evidenced that lower CD, higher VH and VH/CD reflect superior intestinal integrity, which can improve nutrient digestion and absorption, increase disease resistance, and finally promote growth performance [29,30]. Xun et al. [31] revealed that different levels of CUR addition to the diets effectively improved jejunal morphology and the barrier function of weaned piglets. In the present study, it was observed that the supplementation of PE, CUR, and their combination, also promote intestinal morphology. The PE addition showed a superior effect in improving the jejunal morphology compared with duodenal and ileal morphology, indicating that PE supplementation exhibited more beneficial effects on the middle segment of the small intestine.

The physical barrier consists of columnar epithelial cells and the tight junctions between cells (including claudins, occludin and ZO) [32]. Mucins are secreted by a variety of epithelial cells and have protective and renewal effects on epithelial cells. Mucin-2 is a secreted mucin that protects intestinal epithelial cells from pathogenic bacteria, thus acting as an intestinal barrier [33]. When the tight junctions are disrupted, the permeability of the intestine increases, and antigens can pass through the intestinal barrier and jeopardize intestinal health [1]. Fermented-PE treatment prevented the disruption of the architecture of ZO-1 and occludin [7]. Pretreatment with CUR prevents this LPS-induced disorganization of ZO-1, claudin-1, claudin-7, and actin filaments [34]. Previous research had verified the increase in tight junction protein mRNA in mice and jejunal mucosal occludin and ZO-1 mRNA and protein levels in ducks after CUR supplementation [35,36], indicating that CUR enhanced the integrity of the intestinal epithelium. In the present study, CUR supplementation had no significant effect on the tight junction proteins in the jejunum, probably because CUR is not absorbed and utilized in the jejunum, which is consistent with the findings of intestinal morphology in the small intestine showing that CUR increased VH/CD in the duodenum but not in the jejunum. Li et al. [37] found that PR treatment significantly upregulated the relative mRNA levels of intestinal mucin-2 and mucin-4, increased the number of goblet cells, regulated intestinal flora structure, improved the intestinal barrier function, and promoted intestinal health. Consistent with the study, we observed a significant upregulation of mucin-2 gene expression in the jejunum after PE supplementation, which was consistent with the finding that PE supplementation improved intestinal morphology in the jejunum.

## 5. Conclusions

In conclusion, dietary supplementation of PE, CUR, or their combination, did not improve the growth performance of broilers in the 28-day trial, but enhanced the antioxidant status and intestinal integrity of broilers by increasing the activities of antioxidant enzymes and promoting intestinal morphology. The lower doses or encapsulation of PE and CUR might benefit the growth performance of broilers, and they might be effective for challenged birds; both cases need further investigation.

## Figures and Tables

**Figure 1 animals-13-01276-f001:**
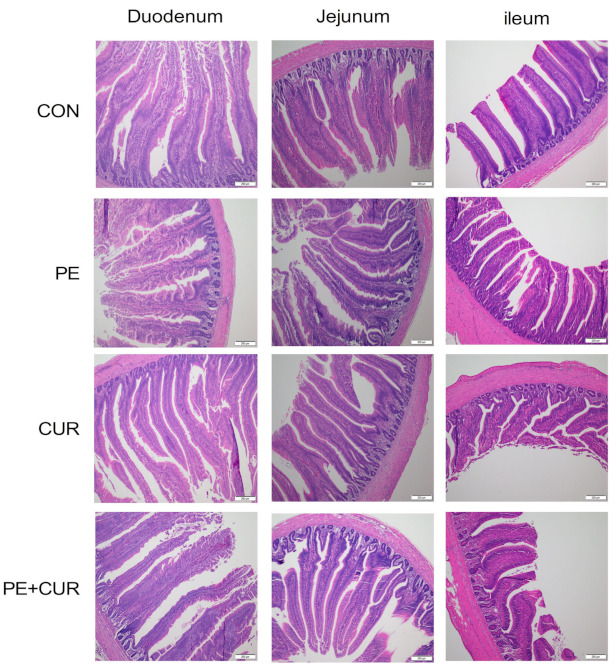
The images of intestinal morphology. CON: control; PE: pueraria extract; CUR: curcumin. Each image is magnified 100×.

**Table 1 animals-13-01276-t001:** Ingredients and nutrient composition of basal diets.

Items	1–28 Days of Age
Ingredients (g/kg of feed)	
Corn	517.3
Soybean meal (crude protein, 43.0%)	407.3
Soybean oil	33.6
Dicalcium phosphate	19.2
Limestone	11.6
Sodium chloride	3.5
DL-Methionine	2.6
Choline chloride (50%)	2.5
Multi-Minerals ^1^	2.0
Multi-Vitamins ^2^	0.4
Calculated nutrient levels (g/kg of dry matter)	
Metabolizable energy (Mcal/kg)	2.92
Dry matter	926.8
Crude protein	215.0
Starch	380.2
Crude fat	57.8
Crude ash	74.5
Calcium	10.0
Available phosphorus	4.5
Lysine	11.7
Methionine + cysteine	9.0
Methionine	5.7
Threonine	8.2

^1^ Multi-Minerals provided per kilogram of diet: Fe, 80 mg; Cu, 8 mg; Mn, 100 mg; Zn, 75 mg; I, 0.35 mg; Se, 0.15 mg. ^2^: Multi-Vitamins provided per kilogram of diet: vitamin A, 12,500 IU; vitamin D_3_, 2500 IU; vitamin E, 30 IU; vitamin K_3_, 2.65 mg; vitamin B_1_, 2 mg; vitamin B_2_, 2 mg; vitamin B_12_ (cobalamine), 0.025 mg; nicotinamide, 50 mg; pantothenate, 12 mg; biotin, 0.0325 mg; folic acid, 1.25 mg.

**Table 2 animals-13-01276-t002:** Primer sequences of target and housekeeping genes.

Target Genes	GenBank Accession Number	Product Size	Primer Sequences (5′-3′)
β-actin	NM_205518.1	152	F:GAGAAATTGTGCGTGATCA R:CCTGAACCTCTCATTGCCA
ZO-1	XM_413773	131	F:CTTCAGGTGTTTCTCTTCCTCCTC R:CTGTGGTTTCATGGCTGGATC
Occludin	D21837.1	123	F:ACGGCAGCACCTACCTCAA R: GGGCGAAGAAGCAGATGAG
Claudin-1	AY750897.1	100	F:CATACTCCTGGGTCTGGTTGGT R:GACACGCATCCGCATCTTCT
Mucin-2	XM_421035	93	F:TTCATGATGCCTGCTCTTGTG R:CCTGAGCCTTGGTACATTCTTGT

ZO-1: zonula occludens-1.

**Table 3 animals-13-01276-t003:** Effects of dietary supplementation with PE and CUR on growth performance of broilers ^1^.

Items	CON	PE	CUR	PE + CUR	SEM	*p*-Value		
						PE	CUR	PE × CUR
1–14 days of age								
ADG (g/d)	24.90	22.38	23.24	22.37	0.34	0.005	0.131	0.136
ADFI (g/d)	30.17	29.99	30.55	30.50	0.41	0.899	0.626	0.943
FCR	1.21	1.34	1.32	1.37	0.03	0.095	0.219	0.521
15–28 days of age								
ADG (g/d)	56.69	53.54	55.81	54.86	0.45	0.019	0.783	0.183
ADFI (g/d)	87.24 ^b^	97.61 ^a^	96.48 ^a^	84.08 ^b^	1.94	0.721	0.453	0.001
FCR	1.56 ^b^	1.81 ^a^	1.75 ^a^	1.54 ^b^	0.04	0.638	0.354	<0.001
1–28 days of age								
ADG (g/d)	40.79	37.96	39.52	38.61	0.33	0.001	0.540	0.068
ADFI (g/d)	58.82 ^ab^	63.58 ^a^	63.47 ^a^	58.00 ^b^	0.95	0.765	0.818	0.005
FCR	1.46 ^c^	1.67 ^a^	1.61 ^ab^	1.51 ^bc^	0.03	1.000	0.142	0.001

^a,b,c^ Means labeled with no common superscript in the same row differed significantly (*p* < 0.05). ^1^ Data were presented with the means and pooled SEM (n = 5). CON: control; PE: pueraria extract; CUR: curcumin; ADG: average daily gain; ADFI: average daily feed intake; FCR: feed conversion ratio.

**Table 4 animals-13-01276-t004:** Effects of dietary supplementation with PE and CUR on antioxidant status in plasma and liver of broilers ^1^.

Items	CON	PE	CUR	PE + CUR	SEM	*p*-Value		
						PE	CUR	PE × CUR
Plasma								
T-AOC (mmol/mL)	1.22	1.15	1.18	1.17	0.01	0.066	0.413	0.100
T-SOD (U/mL)	105.12	110.68	110.41	108.99	1.02	0.306	0.371	0.088
GSH-Px (U/mL)	1150.98 ^d^	2039.73 ^a^	1762.25 ^b^	1293.62 ^c^	63.73	<0.001	0.127	<0.001
CAT (U/mL)	2.27 ^a^	1.61 ^b^	1.43 ^b^	1.81 ^ab^	0.10	0.453	0.094	0.008
H_2_O_2_ (mmol/L)	14.77	12.22	10.50	11.86	0.66	0.648	0.080	0.137
MDA (nmol/mL)	1.44	1.52	1.29	1.36	0.06	0.559	0.198	0.987
Liver								
T-AOC (mmol/mg prot)	0.13	0.14	0.14	0.14	0.00	0.231	0.987	0.379
T-SOD (U/mg prot)	119.77	125.11	119.46	122.30	2.10	0.354	0.722	0.775
GSH-Px (U/mg prot)	55.78	51.41	53.94	44.38	1.22	0.002	0.036	0.211
CAT (U/mg prot)	12.81	10.71	12.11	8.37	0.75	0.052	0.300	0.573
H_2_O_2_ (mmol/g prot)	27.76	28.72	31.12	29.60	0.72	0.847	0.151	0.398
MDA (nmol/mg prot)	1.31	0.87	1.07	1.03	0.06	0.052	0.725	0.105

^a–d^ Means labeled with no common superscript in the same row differed significantly (*p* < 0.05). ^1^ Values are expressed as means and pooled SEM (n = 10). CON: control; PE: pueraria extract; CUR: curcumin; T-AOC: total antioxidant capacity; T-SOD: total superoxide dismutase; GSH-Px: glutathione peroxidase; CAT: catalase; MDA: malondialdehyde.

**Table 5 animals-13-01276-t005:** Effects of dietary supplementation with PE and CUR on antioxidant status in small intestine of broilers ^1^.

Items	CON	PE	CUR	PE + CUR	SEM	*p*-Value		
						PE	CUR	PE × CUR
Duodenum								
T-AOC (mmol/mg prot)	0.35	0.35	0.36	0.39	0.01	0.343	0.140	0.552
T-SOD (U/mg prot)	326.61	315.45	360.62	343.27	6.63	0.262	0.019	0.806
GSH-Px (U/mg prot)	24.47 ^b^	53.19 ^a^	46.05 ^a^	42.86 ^a^	2.78	0.006	0.197	0.001
CAT (U/mg prot)	11.85	13.71	11.72	12.38	0.55	0.270	0.520	0.595
H_2_O_2_ (mmol/g prot)	20.26 ^a^	20.46 ^a^	20.26 ^a^	16.11 ^b^	0.58	0.060	0.040	0.040
MDA (nmol/mg prot)	0.43	0.39	0.40	0.50	0.03	0.671	0.573	0.343
Jejunum								
T-AOC (mmol/mg prot)	0.21	0.21	0.21	0.23	0.01	0.472	0.136	0.310
T-SOD (U/mg prot)	350.95	363.75	353.42	347.16	6.40	0.806	0.597	0.476
GSH-Px (U/mg prot)	27.79	30.71	31.78	30.31	1.18	0.766	0.462	0.368
CAT (U/mg prot)	3.59	3.24	3.40	3.24	0.25	0.630	0.864	0.864
H_2_O_2_ (mmol/g prot)	28.01	29.47	23.90	31.41	1.36	0.103	0.687	0.266
MDA (nmol/mg prot)	1.60	1.46	2.72	1.55	0.25	0.192	0.230	0.302
Ileum								
T-AOC (mmol/mg prot)	0.22	0.24	0.20	0.23	0.04	0.118	0.397	0.725
T-SOD (U/mg prot)	224.66	243.36	246.20	241.20	4.40	0.438	0.274	0.183
GSH-Px (U/mg prot)	37.85 ^b^	41.51 ^ab^	44.69 ^a^	40.83 ^ab^	0.84	0.947	0.052	0.019
CAT (U/mg prot)	2.32	3.15	2.52	2.76	0.13	0.032	0.704	0.230
H_2_O_2_ (mmol/g prot)	16.76	17.97	16.51	16.19	0.82	0.799	0.556	0.658
MDA (nmol/mg prot)	1.04	1.24	1.16	1.05	0.07	0.765	0.845	0.338

^a,b^ Means labeled with no common superscript in the same row differed significantly (*p* < 0.05). ^1^ Values are expressed as means and pooled SEM (n = 10). CON: control; PE: pueraria extract; CUR: curcumin; T-AOC: total antioxidant capacity; T-SOD: total superoxide dismutase; GSH-Px: glutathione peroxidase; CAT: catalase; MDA: malondialdehyde.

**Table 6 animals-13-01276-t006:** Effects of dietary supplementation with PE and CUR on intestinal morphology of broilers ^1^.

Items	CON	PE	CUR	PE + CUR		*p*-Value		
						PE	CUR	PE × CUR
Duodenum								
VH (μm)	1346.63	1189.71	1324.40	1316.70	22.77	0.060	0.222	0.086
CD (μm)	137.67	136.78	123.04	123.80	3.90	0.994	0.085	0.917
VH/CD	9.98	8.86	11.53	10.97	0.33	0.168	0.004	0.643
VA (μm^2^)	251,630.12	239,573.53	253,417.31	232,844.37	6753.94	0.246	0.859	0.760
Jejunum								
VH (μm)	706.37	816.88	827.64	750.62	26.06	0.748	0.598	0.079
CD (μm)	109.07	85.14	103.38	94.23	2.57	0.001	0.695	0.095
VH/CD	6.63 ^b^	9.72 ^a^	8.51 ^ab^	7.49 ^b^	0.38	0.138	0.803	0.005
VA (μm^2^)	106,046.58	156,371.87	112,539.98	116,809.34	6462.33	0.023	0.158	0.053
Ileum								
VH (μm)	633.83	559.72	632.38	682.83	18.32	0.738	0.092	0.085
CD (μm)	84.60	72.46	78.07	82.75	2.50	0.459	0.708	0.100
VH/CD	7.36	7.88	8.02	8.53	0.25	0.302	0.197	0.995
VA (μm^2^)	94,809.34	86,614.06	106,961.24	103,019.27	6820.49	0.667	0.315	0.880

^a,b^ Means labeled with no common superscript in the same row differed significantly (*p* < 0.05). ^1^ Values are expressed as means and pooled SEM (n = 10). CON: control; PE: pueraria extract; CUR: curcumin; VH: villus height; CD: crypt depth; VH/CD: the VH-to-CD ratio; VA: villus area.

**Table 7 animals-13-01276-t007:** Effects of dietary supplementation with PE and CUR on the mRNA levels of claudin-1, occludin, ZO-1 and mucin-2 in jejunum of broilers ^1^.

Items	CON	PE	CUR	PE + CUR		*p*-Value		
						PE	CUR	PE × CUR
ZO-1	1.06	1.03	1.33	1.01	0.05	0.078	0.182	0.139
Occludin	0.98	0.90	0.69	0.88	0.06	0.655	0.191	0.240
Claudin-1	1.08	1.28	1.40	1.32	0.06	0.597	0.138	0.236
Mucin-2	0.96	1.13	0.76	1.07	0.04	0.002	0.074	0.341

^1^ Values are expressed as means and pooled SEM (n = 10). CON: control; PE: pueraria extract; CUR: curcumin; ZO-1: zonula occludens-1.

## Data Availability

Data are contained within the article.

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
