# Peer review of "Effects of Pueraria Extract and Curcumin on Growth Performance, Antioxidant Status and Intestinal Integrity of Broiler Chickens"

_animals, 2023, doi:10.3390/ani13081276_

Round 1
Reviewer 1 Report
Dear authors,
Overall the Introduction chapter is very comprehensive and it describes well the dietary plants to be experimented as well as the objective of the study.
Materials and methods
Row 83-84: For the provenience of two experimental ingredients, please add the countries of those companies you mentioned within text.
Row 85-86: Please mention if room temperature was controlled automatically or manual?
Row 96 : Please mention within brackets the average weight!
Row 106: Eliminate the extra dot
Row 202: I would suggest to add images with VD and CD from different parts of intestinal morphology, and afterword interpret it! I think would be relevant for the manuscript!
Row 216: Please add genotype AA ….so everyone can understand, eventually add more characteristics of this genotype
Row 213: I would like for you to add more references on productive performances on broilers with regards to the 2 ingredients utilization (same observation on GSH-Px, T-SOD, MDA effects ???)
Row 226: In vivo (row 229, also) and in vitro….use italics please!
Overall the Introduction chapter is very comprehensive and it describes well the dietary plants to be experimented as well as the objective of the study.
Materials and methods
Row 83-84: For the provenience of two experimental ingredients, please add the countries of those companies you mentioned within text.
Row 85-86: Please mention if room temperature was controlled automatically or manual?
Row 96 : Please mention within brackets the average weight!
Row 106: Eliminate the extra dot
Row 202: I would suggest to add images with VD and CD from different parts of intestinal morphology, and afterword interpret it! I think would be relevant for the manuscript!
Row 216: Please add genotype AA ….so everyone can understand, eventually add more characteristics of this genotype
Row 213: I would like for you to add more references on productive performances on broilers with regards to the 2 ingredients utilization (same observation on GSH-Px, T-SOD, MDA effects ???)
Row 226: In vivo (row 229, also) and in vitro….use italics please!
Author Response
Dear authors,
Overall the Introduction chapter is very comprehensive and it describes well the dietary plants to be experimented as well as the objective of the study.
Materials and methods
Row 83-84: For the provenience of two experimental ingredients, please add the countries of those companies you mentioned within text.
Answer: The country of companies is added.
Row 85-86: Please mention if room temperature was controlled automatically or manual?
Answer: Room temperature was controlled manually. This information is added.
Row 96 : Please mention within brackets the average weight!
Answer: The average weight is added.
Row 106: Eliminate the extra dot
Answer: The extra dot is eliminated.
Row 202: I would suggest to add images with VD and CD from different parts of intestinal morphology, and afterword interpret it! I think would be relevant for the manuscript!
Answer: The images of intestinal morphology were added in figure 1.
Row 216: Please add genotype AA ….so everyone can understand, eventually add more characteristics of this genotype
Answer: Genotype is added.
Row 213: I would like for you to add more references on productive performances on broilers with regards to the 2 ingredients utilization (same observation on GSH-Px, T-SOD, MDA effects ???)
Answer:Two references was added in productive performances. One reference was added in antioxidant status.
Row 226: In vivo (row 229, also) and in vitro….use italics please!
Answer: The italics of in vivo and in vitro have been revised.
Reviewer 2 Report
Dear Authors
Regarding the manuscript title:
Effects of pueraria extract and curcumin on growth performance, antioxidant status and intestinal integrity of broiler chickens
The scientific background of the topic was well mentioned in the introduction part. The experiment design quite good, as well as the replicates and methods used, were quite good. The results obtained were presented in tables well discussed with other author’s results. However, there are some observation in the present paper should be corrected and add to improve the quality of the paper.
Introduction
Need more information and references about using herbal and its extract in poultry diets as alternative to antibiotics.
Table 1. Ingredients and nutrient composition of basal diets, are you prepared the diets according to Ross 308 requirements?
Add the crude protein of Soybean meal in the table.
At Calculated nutrient level (%) Add Methionine level.
Table 2. Primer sequences of target and housekeeping genes add (Ref. Seq. Accession No. and Annealing for all genes)
Author Response
Dear Authors
Regarding the manuscript title:
Effects of pueraria extract and curcumin on growth performance, antioxidant status and intestinal integrity of broiler chickens
The scientific background of the topic was well mentioned in the introduction part. The experiment design quite good, as well as the replicates and methods used, were quite good. The results obtained were presented in tables well discussed with other author’s results. However, there are some observation in the present paper should be corrected and add to improve the quality of the paper.
Introduction
Need more information and references about using herbal and its extract in poultry diets as alternative to antibiotics.
Answer: Three references have been added in introduction.
Table 1. Ingredients and nutrient composition of basal diets, are you prepared the diets according to Ross 308 requirements?
Answer: As described in the manuscript, the basal diet was formulated according to the NY/T33-2004 recommendations for broilers. The nutrient composition can meet the Ross 308 requirements.
Add the crude protein of Soybean meal in the table.
Answer: This information has been added in Table 1.
At Calculated nutrient level (%) Add Methionine level.
Answer: The methionine level was added.
Table 2. Primer sequences of target and housekeeping genes add (Ref. Seq. Accession No. and Annealing for all genes)
Answer: The accession No. has been presented. Annealing temperature for all genes is 60°C. This information is in the description of RT-PCR.
Reviewer 3 Report
Materials and Methods:
p.2/10, line 83: The concentration of puerarin should be 238g/kg.
p.2/10, Table 1: The ingredients should be expressed in g/kg of feed ore dry matter.
p.3/10, Table 1: The calculated nutrient level should be expressed in g/kg besides Metabolizable energy.
p.3/10, Table 1: As for the nutrient level the contents of dry matter, fat, starch and ash are required.
p.3/10, line 98: The parameters of centrifugation have to be expressed as a/ the diameter of rotor and speed of rotor as rpm or b/ relative centrifugal force (RCF).
p.3/10, line 100: The cervical dislocation without anaesthetic is not active euthanasia.
p.3/10, lines 101-103: A mixture of contradictory information is mentioned in the case of the fixation and storing of 3 parts of the small intestine: a/ storing at -80 oC until analyse (how long?), b/fixation in 4% paraformaldehyde. What is the correct information? It needs explanation.
..
Results:
p.4/10, Table 3: How is it possible that one control and 3 experimental groups have the same SEM value in all lines? Each group of birds should have its own SEM values. The P values on the right side do not correspond with the demonstrated significance on the left side of Table 3 as follows
a/ period 1-14 days of age / ADG:
P=0.005, CON 24.90 vs. PE 22.38 g/kg – P is describing the significance but the values are not significant,
P=0.136, CON 24.90 vs. PE+CUR 22.37 g/kg – P is very high but the value 22.37 is similar to the previous 22.38.
b/ period 15-28 days of age / ADFI:
P=0.001, CON 87.24b vs. PE+CUR 84.08b g/d – P is describing the significance but the values are not significant.
c/ period 15-28 days of age / FCR:
P lower than 0.001, CON 1.56b vs. PE+CUR 1.54b – P is describing the significance but the values are similar.
c/ period 1-28 days of age / ADFI:
P=0.005, CON 58.82ab vs. PE+CUR 58.00b g/d – P is describing the significance but the values are similar.
d/ period 1-28 days of age / FCR:
P=0.001, CON 1.46c vs. PE+CUR 1.51bc – P is describing the significance but the values are similar.
p.5/10, Table 4: The same SEM value for all groups in all lines again? Each group of birds should have its own SEM values. The P values on the right side do not correspond with the demonstrated significance on the left side of Table 4 as follows
a/ plasma / CAT:
P=0.008, CON 2.27a vs. PE+CUR 1.81ab U/ml – P is describing the significance but the values are not significant.
b/ liver / GSH-Px:
P=0.002, CON 55.78 vs. PE 51.41 U/ml – P is describing the significance but the values are not significant.
p.6/10, Table 5: The same SEM value for all groups in all lines again? Each group of birds should have its own SEM values. The P values on the right side do not correspond with the demonstrated significance on the left side of the Table 5 as follows
a/ duodenum / H2O2:
P=0.040, CON 20.26a vs. CUR 20.26a mmol/g prot – P describes the signicance and the values are not significant whereas
P=0.040, CON 20.26a vs. PE+CUR 16.11b mmol/g prot – P is describing the significance and the values different in comaparison to CON-CUR.
b/ ileum / GSH-Px:
P=0.019, CON 37.85b vs. PE+CURab 40.83ab U/ml – P is describing the significance but the values are not significant.
c/ ileum / CAT:
P=0.0032, CON 2.32 vs. PE 3.15 U/mg prot – P is describes the significance but the values are not significant.
p.7/10, Table 6: The same SEM value for all groups in all lines again? Each group of birds should have its own SEM values. The P values on the right side do not correspond with the demonstrated significance on the left side of Table 6 as follows
a/ duodenum / VH/CD:
P=0.004, CON 9.98 vs. CUR 11.53 – P is describing the significance and the values are not significant.
b/ jejunum / CD:
P=0.001, CON 109.07 vs. PE 85.14 – P is describing the significance and the values should be significant as well.
c/ jejunum / VH/CD:
P=0.005, CON 6.63b vs. PE 7.49b – P is describing the significance and the values are not significant?
d/ jejunum / VA:
P=0.023, CON 106046.58 vs. PE 15637187 – P is describing the significance and the values are not significant?
p.7/10, Table 7: The same SEM value for all groups in all lines again? Each group of birds should have its SEM values. The P values on the right side do not correspond with the demonstrated significance on the left side of Table 7 as follows
a/ duodenum / Mucin-2:
P=0.002, CON 0.96 vs. PE 1.13 – P is describing the significance and the values are not significant?
The mentioned data have to be corrected and used in the description of results and the discussion.
Author Response
Materials and Methods:
p.2/10, line 83: The concentration of puerarin should be 238g/kg.
Answer: This has been revised as reviewer suggested.
p.2/10, Table 1: The ingredients should be expressed in g/kg of feed ore dry matter.
Answer: This has been revised as reviewer suggested in Table 1.
p.3/10, Table 1: The calculated nutrient level should be expressed in g/kg besides Metabolizable energy.
Answer: This has been revised as reviewer suggested in Table 1.
p.3/10, Table 1: As for the nutrient level the contents of dry matter, fat, starch and ash are required.
Answer: The information is added in Table 1.
p.3/10, line 98: The parameters of centrifugation have to be expressed as a/ the diameter of rotor and speed of rotor as rpm or b/ relative centrifugal force (RCF).
Answer: The parameters of centrifugation were expressed as relative centrifugal force (RCF).
p.3/10, line 100: The cervical dislocation without anaesthetic is not active euthanasia.
Answer: Selected chickens were intravenously injected with sodium pentobarbital at 50 mg/kg of body weight, and then euthanized by cervical dislocation for slaughter. This information has been added.
p.3/10, lines 101-103: A mixture of contradictory information is mentioned in the case of the fixation and storing of 3 parts of the small intestine: a/ storing at -80 oC until analyse (how long?), b/fixation in 4% paraformaldehyde. What is the correct information? It needs explanation.
Answer: Approximately 1 cm of segments were taken from the middle portion of duodenum, jejunum and, and fixed in 4% paraformaldehyde for morphological analysis. Remaining intestinal samples from duodenum, jejunum and ileum were collected and stored at -80°C until analysis. This information has been added.
Results:
p.4/10, Table 3: How is it possible that one control and 3 experimental groups have the same SEM value in all lines? Each group of birds should have its own SEM values. The P values on the right side do not correspond with the demonstrated significance on the left side of Table 3 as follows.
Answer: SEM is the standard error of mean. It is generally used in all data format, especially in multi-factorial experimental design. The current experiment is 2×2 factorial design. The P values on the right side are main effects of PE and CUR as well as interactive effects between PE and CUR. They are not the simple effect of PE or CUR.
a/ period 1-14 days of age / ADG:
P=0.005, CON 24.90 vs. PE 22.38 g/kg – P is describing the significance but the values are not significant,
Answer: This is the main effect of PE. The data of CON and CUR groups were compared with the data of PE and PE+CUR groups.
P=0.136, CON 24.90 vs. PE+CUR 22.37 g/kg – P is very high but the value 22.37 is similar to the previous 22.38.
Answer: P=0.136 indicates that there is no interactive effects between PE and CUR.
b/ period 15-28 days of age / ADFI:
P=0.001, CON 87.24b vs. PE+CUR 84.08b g/d – P is describing the significance but the values are not significant.
Answer: This means there are significant interactive effects between PE and CUR. When the birds were not fed CUR, PE increased ADFI. When the birds were fed CR, PE decreased ADFI.
c/ period 15-28 days of age / FCR:
P lower than 0.001, CON 1.56b vs. PE+CUR 1.54b – P is describing the significance but the values are similar.
Answer: This means there are significant interactive effects between PE and CUR. When the birds were not fed CUR, PE increased FCR. When the birds were fed CR, PE decreased FCR.
c/ period 1-28 days of age / ADFI:
P=0.005, CON 58.82ab vs. PE+CUR 58.00b g/d – P is describing the significance but the values are similar.
Answer: This means there are significant interactive effects between PE and CUR. When the birds were not fed CUR, PE increased ADFI. When the birds were fed CR, PE decreased ADFI.
d/ period 1-28 days of age / FCR:
P=0.001, CON 1.46c vs. PE+CUR 1.51bc – P is describing the significance but the values are similar.
Answer: This means there are significant interactive effects between PE and CUR. When the birds were not fed CUR, PE increased FCR. When the birds were fed CR, PE decreased FCR.
p.5/10, Table 4: The same SEM value for all groups in all lines again? Each group of birds should have its own SEM values. The P values on the right side do not correspond with the demonstrated significance on the left side of Table 4 as follows
Answer: The same as previous explanation.
a/ plasma / CAT:
P=0.008, CON 2.27a vs. PE+CUR 1.81ab U/ml – P is describing the significance but the values are not significant.
Answer: This means there are significant interactive effects between PE and CUR. When the birds were not fed CUR, PE decreased CAT. When the birds were fed CR, PE did not significantly affect CAT.
b/ liver / GSH-Px:
P=0.002, CON 55.78 vs. PE 51.41 U/ml – P is describing the significance but the values are not significant.
Answer: This is the main effect of PE. The data of CON and CUR groups were compared with the data of PE and PE+CUR groups.
p.6/10, Table 5: The same SEM value for all groups in all lines again? Each group of birds should have its own SEM values. The P values on the right side do not correspond with the demonstrated significance on the left side of the Table 5 as follows
Answer: The same as previous explanation.
a/ duodenum / H2O2:
P=0.040, CON 20.26a vs. CUR 20.26a mmol/g prot – P describes the signicance and the values are not significant whereas
Answer: This is the main effect of CUR. The data of CON and PE groups were compared with the data of CUR and PE+CUR groups.
P=0.040, CON 20.26a vs. PE+CUR 16.11b mmol/g prot – P is describing the significance and the values different in comaparison to CON-CUR.
Answer: This means there are significant interactive effects between PE and CUR. When the birds were not fed CUR, PE did not significantly affect H2O2. When the birds were fed CR, PE decreased H2O2.
b/ ileum / GSH-Px:
P=0.019, CON 37.85b vs. PE+CURab 40.83ab U/ml – P is describing the significance but the values are not significant.
Answer: This means there are significant interactive effects between PE and CUR. When the birds were not fed CUR, PE tended to increase GSH-Px. When the birds were fed CR, PE tended to decrease GSH-Px.
c/ ileum / CAT:
P=0.0032, CON 2.32 vs. PE 3.15 U/mg prot – P is describes the significance but the values are not significant.
Answer: This is the main effect of PE. The data of CON and CUR groups were compared with the data of PE and PE+CUR groups.
p.7/10, Table 6: The same SEM value for all groups in all lines again? Each group of birds should have its own SEM values. The P values on the right side do not correspond with the demonstrated significance on the left side of Table 6 as follows
Answer: The same as previous explanation.
a/ duodenum / VH/CD:
P=0.004, CON 9.98 vs. CUR 11.53 – P is describing the significance and the values are not significant.
Answer: This is the main effect of CUR. The data of CON and PE groups were compared with the data of CUR and PE+CUR groups.
b/ jejunum / CD:
P=0.001, CON 109.07 vs. PE 85.14 – P is describing the significance and the values should be significant as well.
Answer: This is the main effect of PE. The data of CON and CUR groups were compared with the data of PE and PE+CUR groups.
c/ jejunum / VH/CD:
P=0.005, CON 6.63b vs. PE 7.49b – P is describing the significance and the values are not significant?
Answer: This means there are significant interactive effects between PE and CUR. When the birds were not fed CUR, PE increase VH/CD. When the birds were fed CR, PE decreased VH/CD.
d/ jejunum / VA:
P=0.023, CON 106046.58 vs. PE 15637187 – P is describing the significance and the values are not significant?
Answer: This is the main effect of PE. The data of CON and CUR groups were compared with the data of PE and PE+CUR groups.
p.7/10, Table 7: The same SEM value for all groups in all lines again? Each group of birds should have its SEM values. The P values on the right side do not correspond with the demonstrated significance on the left side of Table 7 as follows
Answer: The same as previous explanation.
a/ duodenum / Mucin-2:
P=0.002, CON 0.96 vs. PE 1.13 – P is describing the significance and the values are not significant?
Answer: This is the main effect of PE. The data of CON and CUR groups were compared with the data of PE and PE+CUR groups.
The mentioned data have to be corrected and used in the description of results and the discussion.
Answer: The explanation of results has bee done above. The data are well discussed in the manuscript.
Reviewer 4 Report
Please see comments on attached PDF

Author Response
Simple Summary
Be safe to what?
Answer: Line 13, be safe for animals.
Abstract
Answer: Line 28, the experiment design has bee revised as reviewer suggested.
Introduction
- Line 47, banned is not suitable. It is revised as reviewer suggested.
- Line 48, delete one of.
- Line 49, Be safe for animals.
Materials and Methods
- Lines 74-75, in the experimental design, the main effects and levels were added.
- Line 96, pen average body weight is added. But the birds were not separated weighed with sex (males vs females).
- Line 97, delete randomly.
- Line 100, the three sections were described in detail.
- Line 107, the FCR is mortality adjusted.
- Line 125, the information is added.
Discussion
The advice for future research is added in conclusion.
Round 2
Reviewer 3 Report
I am satisfied with the changes made in the manuscript.
Author Response
Thank you for your feedback.